# HPV-Mediated Resistance to TNF and TRAIL Is Characterized by Global Alterations in Apoptosis Regulatory Factors, Dysregulation of Death Receptors, and Induction of ROS/RNS

**DOI:** 10.3390/ijms20010198

**Published:** 2019-01-08

**Authors:** Tatiane Karen Cabeça, Alice de Mello Abreu, Rafael Andrette, Vanesca de Souza Lino, Mirian Galliote Morale, Francisco Aguayo, Lara Termini, Luisa Lina Villa, Ana Paula Lepique, Enrique Boccardo

**Affiliations:** 1Department of Microbiology, Institute of Biomedical Sciences, University of São Paulo, 05508-900 São Paulo, Brazil; taticabeca@yahoo.com.br (T.K.C.); liabreu@yahoo.com.br (A.d.M.A.); rafaelandrette@outlook.com (R.A.); vanesca_lino@hotmail.com (V.d.S.L.); 2Laboratório de Inovação em Câncer, Centro de Investigação Translacional em Oncologia (LIM24), Instituto do Câncer do Estado de São Paulo (ICESP), 01246-000 São Paulo, Brazil; mirian.galliote@gmail.com (M.G.M.); terminilara@gmail.com (L.T.); luisapvilla@gmail.com (L.L.V.); 3Departamento de Radiologia e Oncologia, Faculdade de Medicina da Universidade de São Paulo (USP), 01246-903 São Paulo, Brazil; 4Department of Basic and Clinical Oncology, Faculty of Medicine, University of Chile, 8389100 Santiago, Chile; faguayo@med.uchile.cl; 5Advanced Center for Chronic Diseases (ACCDiS), Universidad de Chile, 8389100 Santiago, Chile; 6Department of Immunology, Institute of Biomedical Sciences, University of São Paulo, 05508-900 São Paulo, Brazil; alepique@icb.usp.br

**Keywords:** HPV, TNF, TRAIL, apoptosis, ROS/RNS

## Abstract

Persistent infection with high-risk human papilloma virus (HR-HPV) is the main risk factor for the development of invasive cervical cancer although is not sufficient to cause cervical cancer. Several host and environmental factors play a key role in cancer initiation/progression, including cytokines and other immune-response mediators. Here, we characterized the response to the individual and combined action of the pro-inflammatory cytokines tumor necrosis factor (TNF) and TNF-related apoptosis-inducing ligand (TRAIL) on HPV-transformed cells and human keratinocytes ectopically expressing E6 and E7 early proteins from different HPV types. We showed that keratinocytes expressing HPV early proteins exhibited global alterations in the expression of proteins involved in apoptosis regulation/execution, including TNF and TRAIL receptors. Besides, we provided evidence that TNF receptor 1 (TNFR1) was down-regulated and may be retained in the cytoplasm of keratinocytes expressing HPV16 oncoproteins. Finally, fluorescence analysis demonstrated that cytokine treatment induced the production and release of reactive oxygen and nitrogen species (ROS/RNS) in cells expressing HPV oncogenes. Alterations in ROS/RNS production and apoptosis regulatory factors expression in response to inflammatory mediators may favor the accumulation of genetic alterations in HPV-infected cells. Altogether, our results suggested that these events may contribute to lesion progression and cancer onset.

## 1. Introduction

Human papilloma viruses (HPVs) are non-enveloped small DNA viruses that infect keratinocytes of the skin and mucosa at different anatomic locations. Persistent infection with a subset of HPV types, collectively known as high-oncogenic risk HPV types (HR-HPV), is the main risk factor for the development of cervical cancer and its precursor lesions, one of the most common cancers in women worldwide [1].

Several inflammatory mediators are produced by host cells in response to viral infections. These pro-inflammatory stimuli trigger different signaling pathways that lead to cell growth arrest or cell death and are critical for determining viral clearance or persistence [2]. Tumor necrosis factor-α (TNF) and TNF-Related Apoptosis-Inducing ligand (TRAIL) are two important mediators of skin and mucosa inflammation [3,4]. These cytokines exert a potent cytostatic effect on keratinocytes in different culture systems [5,6,7]. TNF can bind to two structurally distinct receptors present on the membrane of target cells, namely TNFR1 (p55/TNFRSF1A) and TNFR2 (p75/TNFRSF1B). Binding of TNF to the cysteine-rich extracellular domains of TNFR1 induces receptor trimerization leading to physical approximation of death-domains present in the cytoplasmic portion of the protein. Depending on the cell type and the cellular microenvironment, oligomerization of these proteins may result in caspase activation and apoptosis [8]. Alternatively, TNFR1 activation may induce NFκB-dependent and independent protective responses [9,10]. In humans, TRAIL may bind to four membrane receptors. Specifically, TRAILR1 (TNFRSF10A) and TRAILR2 (TNFRSF10B) are pro-apoptotic receptors which mediate the formation of a death-inducing signaling complex upon TRAIL binding leading to caspase cascade activation and apoptosis. Besides, this cytokine may bind other receptors that act as decoys and prevent death signaling by competing for TRAIL and/or by activating cell survival pathways [8].

To warrant progeny production and spread, viruses have evolved a plethora of mechanisms to evade innate and adaptive immune barriers imposed by the host. As obligatory intracellular parasites, one crucial way used by viruses to avoid elimination is to hamper host-mediated apoptosis induction pathways. This includes prevention of the interaction of ligands of the TNF-superfamily and their cognate receptors, prevention of caspase activation, interference with the formation of the death-inducing signaling complex (DISC) and mitochondria targeting [11,12].

HPV early genes *E6* and *E7* codify for the major viral transforming proteins. These oncoproteins cooperate for the efficient immortalization and transformation of primary human keratinocytes in vitro [13,14]. Besides, their continued expression is critical for maintenance of the transformed phenotype in cervical cancer-derived cell lines [15]. These oncoproteins interact with several cellular factors and interfere with different signaling pathways to induce cell proliferation, escape apoptosis and evade immune surveillance. The main properties of these oncoproteins are reviewed elsewhere [16].

Previously, reported data suggest that acquisition of resistance to TNF and TRAIL is an early event during viral infection and may be required for viral persistence as well as for malignant progression. This notion is supported by the fact that HPV oncoproteins modulate the response to these cytokines in different models [7,17,18,19]. For instance, HPV16 E6 induces resistance to TNF-mediated apoptosis and cell cycle withdrawal in mice fibroblasts and human keratinocytes, respectively [20,21]. Besides, E7 proteins from HPV16 and HPV18 induce resistance to TNF antiproliferative effect on primary human keratinocytes [6,7,21]. In addition, we previously reported that ectopic expression of HPV16 E7 causes major alterations in the global gene expression profile of keratinocytes exposed to TNF [7]. TRAIL expression and activity is another target of HPV early proteins. For instance, TRAIL expression is inhibited in keratinocytes harboring complete HPV16, -18, and -31 genomes [22]. Moreover, HPV16 E5 and E6 early proteins impair TRAIL-mediated apoptosis in human keratinocytes and HCT116 cells, respectively [18,19].

Despite of these observations, the effect of simultaneous TNF and TRAIL interaction on cells expressing HPV E6 and E7 genes has not been investigated. In this study, we analyzed the effect of TNF and TRAIL on the proliferation and viability of cervical cancer-derived cell lines and primary human keratinocytes (PHK) transduced with E6 and E7 of HPV11 and HPV16 grown in monolayer and organotypic cultures. We observed that E6 and E7 from both low- and high-oncogenic risk types conferred resistance to both cytokines when administered alone or in combination. In addition, we demonstrated that these cells exhibited alterations in the global expression of genes involved in NFκB signaling pathway and in apoptosis regulatory/executioner proteins. Besides, we showed that the expression profiles of these factors vary between cells expressing E6 and E7 from low- or high-risk oncogenic types. Using this approach, we identified new HPV-associated alterations in the apoptosis machinery. These included the up-regulation of receptors TRAIL R1/DR4, TRAIL R2/DR5, Fas/TNFSF6/CD95 and TNF R1/TNFRSF1A in cells expressing HPV16 E6 and E7. We provided evidence that TNF R1/TNFRSF1A was retained in the cytoplasm of cells expressing HPV16 oncogenes, hampering receptor activation. Finally, we showed that exposure to pro-inflammatory cytokines induced the production and release of reactive oxygen species (ROS)/reactive nitrogen species (RNS) in HPV-positive cells.

## 2. Results

### 2.1. HPV E6 and E7 Confer Resistance to both TNF and TRAIL

We have previously shown that HPV-transformed cell lines and cells expressing E7 from HR-HPV types are resistant to TNF’s antiproliferative effect [5,7,21,23]. However, the combined effect of TNF and TRAIL has not been addressed. This constitutes an important issue since it is expected that in biological systems different cytokines may act simultaneously on the same cell type. First, we analyzed the effect of TRAIL on cervical cancer-derived cell lines SiHa (HPV16), HeLa (HPV18) and C33 (HPV-negative). Figure 1A shows the results of a representative proliferation assay. We observed that the effect of different concentrations of TRAIL varied between cell lines. For instance, proliferation/viability of SiHa and HeLa cell lines was basically unaffected by TRAIL. On the other hand, the HPV-negative cell line C33 exhibited some degree of sensitivity to TRAIL resulting in a drop-in cell proliferation/viability that, although consistent, varied in magnitude between experiments (Appendix A).

We next determined the combined effect of TNF and TRAIL on cervical cancer-derived cells. The results presented in Figure 1B confirm previous observations since the three transformed cell lines were resistant to TNF antiproliferative effect. Besides, we observed that proliferation of these cells was not affected when TNF and TRAIL were applied simultaneously (Figure 1B and Appendix A). Altogether, these results indicate that cervical cancer-derived cell lines are resistant to the effect of TRAIL on cell proliferation/viability.

Our results suggest that HPV, probably through the action of specific gene products, confers resistance to the effect of TRAIL. To study the role of specific HPV genes in this process we performed similar assays using PHK transduced with retroviral vectors expressing *E6* and *E7* genes of HPV11 (low-oncogenic risk) and HPV16 (high-oncogenic risk). The results are shown in Figure 1C–F. We observed that PHKs transduced with control vectors are clearly sensitive to TRAIL. In these cells, cytokine treatment caused a reduction in cell proliferation/viability that varied between 45% and 60% (Figure 1C,E). On the other hand, keratinocytes expressing E6 and E7 of HPV11 exhibited a lower sensitivity to TRAIL effect with a reduction of cell proliferation/viability around 20% (Figure 1C). Besides, PHKs transduced with HPV16 oncogenes were completely resistant to TRAIL (Figure 1E). These data show that E6 and E7 HPV proteins, mainly those from HR-HPV types confer resistance to TRAIL. Furthermore, we show that cells expressing *E6* and *E7* genes from either low- or high-oncogenic risk were more resistant to the combined action of TNF and TRAIL when compared to control cells (Figure 1D,F). Altogether, our results show that HPV E6 and E7 acutely confer resistance to pro-inflammatory cytokines TNF and/or TRAIL treatment.

### 2.2. HPV16 E6 and E7 Differentially Alter the Expression of Genes Involved in the NFκB Pathway

The observations described above suggested that cervical cancer cell lines as well as PHKs transduced with HPV16 E6 and E7 exhibited perturbations in both TNF and TRAIL-regulated pathways. To test this hypothesis, we decided to analyze the expression of genes pertaining to the NFκB signaling pathway since this transcription factor is activated by both cytokines and regulates many genes, including many involved in anti-viral responses. This study was performed using organotypic epithelial cultures to recapitulate epithelial differentiation program which is fundamental for HPV life cycle. Our results show that 19 out of 84 genes present in the NFκB array were differentially expressed (*fold* > 1.5 or < −1.5; *p*-value < 0.05) between control and HPV16 oncogenes expressing cells. Twelve of these genes were up-regulated and 7 were down-regulated in keratinocytes expressing HPV16 E6 and E7 (Table 1). Among the up-regulated genes, we found genes related to cell survival, from the B-cell lymphoma (BCL) family. Interestingly, genes which products are related to inflammation were both up- and down-regulated. For example, we found that CSF3 was up-regulated, while IL-8 was down-regulated. Both genes display a role in the expansion and recruitment of granulocytes to the tumor microenvironment. We also observed up-regulation of IKBκB, NFκB1, TBK1, and down-regulation of MyD88 and TLR3, all proteins in the signaling pathway regulated by Toll-like receptors (TLRs). These results confirm the existence of important alterations in NFκB pathway in cells expressing HPV16 oncogenes.

### 2.3. HPV11 and HPV16 E6 and E7 Differentially Regulate the Expression of Proteins Involved in Apoptosis Regulation/Execution

Due to the dual role TRAIL, TNF and effector pathways may have on cell fate, and the ambiguous results we obtained regarding the expression of NFκB targets, we next decided to analyze the effect of HPV E6 and E7 early proteins, TNF or TRAIL treatment on the expression of proteins involved in apoptosis regulation/execution. Total protein extracts from epithelial organotypic cultures treated with 2 nM of TNF or 25 ng/mL TRAIL for 72 h were analyzed using the Proteome Profiler™ Array-Human Apoptosis Array Kit (R&D Systems, #ARY009) that includes 35 different proteins. The results obtained are presented in Figure 2 and Appendix A.

We observed differences in the levels of several factors when comparing HPV16 E6 and E7 expressing cultures with controls. For instance, we observed that cultures from cells expressing HPV16 oncoproteins have up-regulated levels of surviving, X-linked inhibitor of apoptosis protein (XIAP), HTRA2/Omi, TRAIL R2, Fas receptor (CD95), TNF R1, p21cip1/waf1 and p27kip1, among others. The proteins p21cip1/waf1 and p27kip1 are cyclin dependent kinases inhibitors and are involved in cell cycle regulation. These results agree with previous studies that showed that these factors are present at high levels in cells expressing HPV oncogenes [24,25].

Analyzing the general expression profile of all the proteins included in the array we observed that several pro-apoptotic factors including Bad, Bax, HTRA2/Omi and activated caspase 3 were up-regulated in cultures expressing HPV16 E6 and E7 proteins. On the other hand, several anti-apoptotic factors such as survivin and XIAP were also up-regulated in these cells. Alterations in the levels of factors involved in apoptosis regulation were also observed in cultures expressing E6 and E7 from HPV11, a low-oncogenic risk HPV type. In this case, we detected an overall down-regulation of factors involved in apoptosis regulation (Appendix A). Our observations strongly suggest that alterations in anti- as well as pro-apoptotic factors induced by low-oncogenic risk HPV types differ from those induced by oncogenic HPV types. Besides, our results confirm and further extend the existence of important alterations in the apoptotic machinery in cells expressing HPV *E6* and *E7* genes.

Importantly, no major alterations were observed in the expression pattern of proteins related to apoptosis regulation upon treatment with TNF or TRAIL (Figure 2B and Appendix A). One exception was the hypoxia inducible factor alpha (HIF1-α). We observed that the levels of this factor were up-regulated in untreated cells expressing HPV16 oncoproteins confirming observations from previous studies (Figure 2A) [26,27,28]. On the other hand, HIF1-α was down-regulated upon treatment with TRAIL (Figure 2B). Analyzing the expression of factors associated with cellular stress we observed that control keratinocytes cultures exhibited higher levels of the CBP/p300-interacting transactivator 2 (CITED2) than cultures expressing HPV16 E6 and E7 (data not shown). CITED2 competes with HIF1-α for binding to p300, inhibiting HIF1-α mediated transactivation of its target genes [29,30]. Besides, it was reported that CITED2 may inhibit NFκB activity by preventing p65 acetylation and, consequently, binding to HIF1-α promoter [31]. This observation suggests that cells expressing HPV16 oncogenes may overcome HIF1-α down-regulation by TRAIL by down-regulating a direct HIF1-α competitor.

### 2.4. HPV16 Oncogenes Regulate TNFR1 Expression in Human Keratinocytes

A second important observation made in our study was that the expression of the receptors TRAIL R1/DR4, TRAIL R2/DR5, Fas/TNFSF6/CD95 and TNF R1/TNFRSF1A were up-regulated in epithelial organotypic cultures expressing HPV16 E6 and E7 (Figure 2; Figure 3). Similar results were obtained after analysis of protein extracts derived from organotypic cultures treated with TNF (Appendix A) or TRAIL (Figure 2) showing that our observations in this model are consistent. These observations were unexpected considering the data described above and previously published studies addressing the resistance of cells expressing HPV oncogenes grown in monolayer to TNF and TRAIL [6,7]. There are at least two possible explanations for this finding which are not mutually exclusive. The first is that although these receptors are up-regulated, some factor, most probably of viral origin, interferes with its mediated signal transduction. Another possibility is that these receptors exhibit alterations in their cellular localization in the presence of viral proteins.

To address this second possibility, we analyzed the expression of TNF receptor 1 (TNFR1) and TRAIL-receptor 1 (TRAIL R1/DR4) by immunofluorescence in keratinocytes expressing HPV16 E6 and E7. No major differences were observed for TRAIL R1/DR4 levels or localization between control and E6 and E7 expressing cells (Figure 4A). On the other hand, the cellular distribution and levels of TNFR1 were clearly affected in the presence of HPV proteins. While control cells exhibited a homogeneous distribution pattern all over the cells with areas of concentration at the cellular membrane, cells expressing HPV16 oncoproteins exhibited lower TNFR1 levels located mainly in the perinuclear area (Figure 4B). TNFR1 down-regulation, in the presence of both HPV16 oncogenes, a condition that reflects what it is observed in natural HPV infections, was confirmed by western blot (Figure 4C). These observations suggest that HR-HPV oncoproteins alter TNFR1 levels and induce receptor relocation in cells grown in monolayer contributing to explain, at least in part, the altered response to TNF exhibited by these cells. Besides, the divergent results obtained in monolayer and organotypic cultures indicate that HPV oncoproteins may regulate TNFR1 expression/activity by different mechanisms in different epithelial strata and differentiation conditions (see Section 3).

### 2.5. Treatment with TNF and/or TRAIL Induces ROS/RNS in Cells Expressing HPV Genes

We have so far demonstrated that treatment with TRAIL and TNF modulate the signaling of pathways involved in response against pathogens, cell death and survival and even metabolism. Oxidative stress (OS), which is related to the processes described above, is considered and important co-factor in HPV-mediated carcinogenesis. Clinical and epidemiological observations as well as biochemical data support the notion that infection with HR-HPV types, the establishment of persistent infection and viral integration are potentiated by OS [32]. Since TNF and TRAIL promote OS as part of their mechanism of action, we studied the production of ROS, RNS and hydrogen peroxide (H_2_O_2_) in cervical cancer-derived cell lines and in keratinocytes transduced with HPV early genes. Intracellular and supernatant ROS/RNS and H_2_O_2_ (not shown) levels were determined using the OxiSelect™ In Vitro ROS/RNS Assay kit (Figure 5). We observed that all cell lines exhibited similar levels of intracellular ROS/RNS levels. On the other hand, control keratinocytes cultures exhibited lower levels of ROS/RNS in cell supernatant compared to cells expressing HPV oncogenes (Figure 5A). We observed that extracellular levels of ROS/RNS increased in response to cytokines in a cell dependent manner (not observed in SiHa) particularly in response to TRAIL. We also observed that single or combined treatment with TNF and/or TRAIL induced a statistically significant increase in the levels of intracellular ROS/RNS species in HR-HPV-positive cell lines, while this was not observed for PHKs in any of the conditions tested (Figure 5, solid lines). On the other hand, we observed that expression of E6 and E7 on keratinocytes led to a significant decrease in the mitochondrial membrane potential. TNF treatment further reduced the mitochondrial membrane potential in E6/E7 positive cells, but not in control cells (Figure 6). These results suggest that the increase in ROS production is independent of the mitochondrial activity. It is possible that, TNF, through the NFκB pathway may lead to NAPDH complex activity to generate ROS.

## 3. Discussion

In the present study, we analyzed the effect of pro-inflammatory cytokines TNF and TRAIL on cells expressing HPV oncogenes. Signaling triggered by these cytokines is complex and depends on the target cell. These cytokines can induce cell death or survival. Cell death is induced through activation of caspases, and survival through activation of the canonical NFκB pathway, and in the case of TNF, also through c-Jun N-terminal kinase (JNK) pathway and AP-1, or in the case of TRAIL, also through mitogen-activated protein kinase (MAPK) and PI3K [33].

Previous studies have addressed the individual effect of these cytokines on cells expressing HPV oncogenes. However, this is the first time that the combined effect of these cytokines is analyzed on HPV-transformed cells and PHKs expressing E6 and E7 from different HPV types. This is a relevant issue since in biological systems several cytokines and other soluble factors may act simultaneously on target cells. In this context, the final cellular response will depend on the integration of signals transduced by several pathways.

We first observed that TRAIL inhibits keratinocytes proliferation in a dose dependent manner. This observation is very interesting since it has been reported that TRAIL does not induce growth arrest in keratinocytes which is in clear contradiction to the results presented here (Figure 1) [6]. A plausible explanation for these contradictory results is that in their study the authors used a non-cross-linked TRAIL form while we used SuperKillerTRAIL™ that uses a peptide to form a more stable oligomer with enhanced ligand activity and higher biological effect. The ability of HPV16 oncoproteins to disturb the response to TRAIL has been described before and highly depends on the experimental model used. For instance, it has been reported that HPV16 E6 can protect HCT116 cells from TRAIL-mediated apoptosis [19], while E7 sensitizes keratinocytes to apoptosis induced by this cytokine in the presence of protein synthesis inhibitors [7]. Here, we show that HPV-transformed SiHa and HeLa cell lines as well as keratinocytes expressing E6 and E7 from both low- and high-oncogenic-risk HPV types are resistant to TRAIL. Besides, we observed that HPV16 proteins are more efficient in conferring resistance to this cytokine than those from HPV11.

We then analyzed the combined effect of TNF and TRAIL on the cells under study. Overall, the three cervical cancer-derived cell lines were resistant to treatment with these cytokines. Again, keratinocytes transduced with HPV11 or HPV16 *E6* and *E7* genes were also resistant to the combined action of TNF and TRAIL. Altogether, our results suggest that resistance to cytokines of the TNF family is a general trait of HPV-infected cells. Furthermore, the fact that cells expressing E6 and E7 proteins from a low-oncogenic risk type are resistant to these cytokines support the notion that overcoming TNF and TRAIL cytostatic effect is needed at the very early stages of viral infection and it is not linked to cell immortalization or abrogation of p53 or pRb functions.

The observations described above show that cells expressing HPV oncogenes exhibit perturbations in both TNF and TRAIL pathways. Therefore, we decided to analyze NFκB signaling pathway since this transcription factor may be activated by both cytokines and regulates a great number of genes [34]. Using a commercial qPCR array, we identified 19 genes related to NFκB signaling pathway which mRNA expression was differentially regulated in organotypic cultures expressing HPV16 E6 and E7 when compared to controls. Previous studies have shown that the E7 protein from HR-HPV types reduced TNF-mediated NFκB activation [35]. The effect of NFκB activation in keratinocytes expressing HPV16 oncoproteins is unknown. Several studies indicate that activated NFκB is a negative regulator of keratinocytes proliferation in the epidermis playing an important role in cell differentiation and tissue homeostasis [36,37,38,39,40,41]. However, it has also been proposed that alterations in NFκB activation or in genes regulated by this factor may play a role in HPV-mediated pathogenesis [42,43]. For instance, it was reported that HPV16 E6 down-regulation by ectopic E2 expression potentiates NFκB activation by TNF in SiHa cells and that this is associated with increased colony formation efficiency [44]. Importantly, our results demonstrate the presence of alterations in the NFκB signaling pathway in organotypic cultures expressing HPV16 oncogenes. These observations support the existence of a complex interaction between HR-HPV oncogenes and TNF/NFκB signal axis relevant to both HPV life cycle and HR-HPV-mediated pathogenesis.

Activation of TNFR1 and TRAIL receptors by their respective ligands approximate their death-domain present at the cytoplasmic face of the membrane. This recruits the death-domain-containing adapter molecules TNFR1-associated death domain protein (TRADD) and Fas-associated protein with death domain (FADD) which in turn transmit a death signal resulting in caspase activation and apoptosis. To better understand the effect of these cytokines on cells expressing HPV early E6 and E7 genes, we analyzed the expression of 35 proteins involved in apoptosis regulation/execution. We observed that several pro-apoptotic, including Bad, Bax, HTRA2/Omi and activated caspase 3, as well as anti-apoptotic factors, including survivin, cIAP1, cIAP2 and XIAP, were up-regulated in organotypic cultures expressing HPV16 E6 and E7 proteins. Previous studies have shown that HPV16 E6 can activate the survivin promoter [45] and induce the cellular apoptosis inhibitor cIAP1 [46]. Induction of cIAP2 has been detected in cells expressing HPV16 E6 and E7 [47]. However, the induction of the X-linked inhibitor of apoptosis protein XIAP by HPV genes has not been reported before. These apparently contradictory results underscore the complex dynamics that exist between HPV infection cells and cell death regulation. Here is important to take into consideration the advantages of using the organotypic culture system for this kind of study. This model recapitulates all the stages of epithelium differentiation which are critical for HPV life cycle allowing the study of biological relevant virus-host cell interactions [48]. This model has been used to demonstrate that HR-HPV E7 is critical to induce cell proliferation in suprabasal layers of the epithelium [49] even in the presence of cytostatic effect imposed by TNF treatment [7,21].

In normal settings, unscheduled cell proliferation/DNA synthesis triggers pro-apoptotic cellular mechanisms to control this effect. In this context, E6 expression and its multiple effects on cell’s death machinery are fundamental for HPV-infected cell survival. In fact, it has been shown that activation of apoptotic pathways is important for HPV replication in upper layers of the epithelium [50]. Besides, resistance to different pro-apoptotic stimuli induced by this protein is important in HPV-mediated pathogenesis [16].

Apoptosis triggering is determined by an imbalance between pro- and anti-apoptotic factors’ expression. Our results, obtained after extensive analysis of apoptosis regulators/executioners proteins underscore the existence of a global equilibrium between pro- and anti-apoptotic factors in cells expressing HPV16 oncogenes. Overall, this equilibrium is not disturbed after treatment with pro-inflammatory cytokines. Therefore, our results strongly support the notion that HPV infection has an important effect on apoptosis machinery to warrant cell survival. We also observed that organotypic cultures expressing E6 and E7 from a low-oncogenic risk HPV type also exhibit variations in the expression profile of proteins involved in apoptosis regulation/execution. However, these alterations are different from those observed in cultures expressing HPV16 E6 and E7 oncoproteins. This observation suggests that global alteration in the apoptotic machinery is a common event in cells expressing HPV genes. However, the nature and magnitude of this change depends on the HPV type analyzed.

One critical observation in our study is that many death-ligand receptors, including TRAIL R1/DR4, TRAIL R2/DR5, Fas/TNFSF6/CD95 e TNF R1/TNFRSF1A are up-regulated in cells expressing HPV16 oncogenes grown in organotypic cultures. Intriguingly, analysis of cells expressing E6 and E7 grown in monolayer showed an opposite result. At present, we do not know the molecular basis and biological relevance of this difference. However, the observation that HPV16 E6 induces a marginal reduction in TNFR1 receptor levels in monolayer PHK cultures parallels the fact that E6 from HPV16 and 18 induces resistance to TNF antiproliferative effect in the basal layer of organotypic cultures [21,51]. We speculate that HPV E6 and E7 proteins may regulate TNFR1 expression/activity by different mechanisms in different epithelial strata. Further investigations are necessary to clarify this issue. All in all, we provide evidence that at least TNFR1 is down-regulated in monolayer cultures of PHK expressing HPV16 oncoproteins. Besides, our results indicate that this receptor may be retained in the cytoplasmic compartment in these cells. We are currently analyzing the cell distribution of other death receptors in HPV-infected cells. However, the observation made for TNFR1 suggests that receptor reallocation may be a novel HPV-mediated mechanism to modulate the cell response to pro-inflammatory cytokines.

Infection and chronic inflammation are driving forces for carcinogenesis. Under inflammatory conditions, ROS and RNS are generated from inflammatory and epithelial cells and result in oxidative and nitrative DNA damage. Here we observed that treatment with TNF or TRAIL induced a statistically significant increase in the intracellular concentration of ROS/RNS in HPV-transformed cells as well as in keratinocytes expressing HPV16 oncoproteins (Figure 5B–D, solid lines). Interestingly, this was not observed in normal PHKs (Figure 5A). Persistent infection by HR-HPV types, which is likely paralleled by persistent inflammation, is the main risk factor for the development of cervical cancer precursor lesions [52].

In conclusion, our results suggest that long term exposure of HPV-infected cells to TNF and/or TRAIL may result in the accumulation of ROS/RNS that may lead to DNA damage. Besides, resistance to cytostatic and pro-apoptotic effect of pro-inflammatory cytokines can cause the propagation of DNA lesions and mutations that may play a role in the initiation and/or promotion of inflammation-mediated carcinogenesis.

## 4. Materials and Methods

### 4.1. Cells, Retroviruses and Organotypic Raft Cultures

Cervical cancer-derived cell lines SiHa (HPV16, ATCC #HTB-35), HeLa (HPV18, ATCC #CCL-2) and C33 (HPV-negative, ATCC #HTB-31) were cultured in MEM (Invitrogen, Carlsbad, CA, USA) supplemented with 10% BCS (Cultilab, Campinas, SP, Brazil) and maintained at 37 °C and 5% CO_2_. Low-passage pooled neonatal foreskin keratinocytes (Lonza Walkersville, Inc., Walkersville, MD, USA) were grown in serum-free medium (Invitrogen, Frederick, MD, USA). At passage one, cells were acutely infected with recombinant pLXSN or pBabe retroviruses expressing the neomycin selection marker. After 24 h, the cells were selected with 300 µg/mL of G418 (pLXSN) or 2 µg/mL puromycin (pBabe) for 2 days, when 100% of mocked infected controls were dead. Surviving cells were amplified and used to seed the epithelial raft cultures without extensive passage or to seed monolayers cultures. Recombinant pBabe retroviral vectors either empty or containing HPV11 E6 and E7 genes were kindly provided by Dennis J. McCance (University of New Mexico, Albuquerque, NM, USA). Recombinant pLXSN-based retroviral vectors for expression of HPV16 oncogenes were a gift from Dr. Denise Galloway (Fred Hutchinson Cancer Research Center, Seattle, WA, USA). These vectors are described elsewhere [53,54].

### 4.2. Treatment with Cytokines and Proliferation Analysis

For cell proliferation analysis, cells were seeded in 96-well plates (2000 cells/well) and after 24 h they were treated with 2 nM TNF (Roche Applied Science, Indianapolis, IN, USA), or with 5, 10, 25, 50 or 100 ng/mL of TRAIL (SuperKillerTRAIL, Enzo Life Sciences, Farmingdale, NY, USA), or with 2 nM TNF plus 25 or 50 ng/mL of TRAIL for 60 h. The TNF concentration used corresponded to the one detected in epithelial lesions and that showed higher antiproliferative effect on keratinocytes as shown by us [5,7,21,23,55]. All treatments were performed in octuplicates. At the end of the treatment, 10 µL of Alamar Blue (Invitrogen, Frederick, MD, USA) were added per well and cells were incubated at 37 °C for 4 h. After this period, Alamar Blue’s reduction was monitored at 570 and 600 nm in an Epoch Microplate Spectrophotometer (Bio-Tek, Winooski, VT, USA). For RNA and protein expression analysis, organotypic cultures maintained for 9 days at the medium-air interface raft cultures were treated with 2 nM TNF or 25 ng/mL of TRAIL for 72 h.

### 4.3. Analysis of the Expression of Genes Involved in NFκB Signaling Pathway

The epithelial component of organotypic cultures was detached from the dermal equivalent using forceps and scalpel. Samples were immediately stored at −20 °C in RNA Later (Qiagen, Hilden, Germany). Total RNA was isolated using TRIzol^®^ (Invitrogen, Frederick, MD, USA) according to the manufacturer’s instructions. RNA integrity was determined by direct visualization of samples electrophoresed in 1% agarose gel in TAE buffer (40 mM Tris-Acetate, 1 mM EDTA). RNA samples were then treated with RQ1 RNase free DNase (M198A, Promega, Madison, WI, USA) according to the manufacturer’s instructions. cDNA was synthetized using the Improm-II™ reverse transcription system (A3800, Promega, Madison, WI, USA) according to the manufacturer’s instructions. Finally, 1 µg of cDNA from each sample was used for RNA expression profile using the Human NFκB Signaling Pathway array (#330231, Qiagen, Hilden, Germany) according to the manufacturer’s instructions in an ABI 7300 equipment (Applied Biosystems, Warrington, UK).

### 4.4. Protein Extraction, Quantification and Expression Analysis

After 12 days at the liquid-air interface (including 72 h of treatment with 2 nM of TNF or 25 ng/mL of TRAIL), raft cultures were detached from the collagen matrix with a scalpel and frozen in liquid nitrogen. Two rafts were minced in a tissue grinder containing lysis buffer from the Proteome Profiler^TM^ Array—Human Apoptosis Array Kit (#ARY009, R&D Systems, Minneapolis, MN, USA), processed and analyzed according to manufacturer’s instructions. Protein concentration was determined using the BCA Protein Assay kit (#23225, Thermo-Scientific, Waltham, MA, USA). Data was analyzed using ImageJ software version 1.51 (https://imagej.nih.gov/ij/).

### 4.5. Immunofluorescence

Control or HPV16 E6E7 expressing keratinocytes were seeded in 8-wells Lab-Tek chambers (Thermo-Scientific, Waltham, MA, USA) at 2 × 10^4^ cells/well. Cells were fixed with a buffered 4% formaldehyde solution for 15 min and permeabilized in methanol for 2 min. Cells were incubated with anti-DR4 or anti-TNFR1 (Abcam, Cambridge, MA, USA) ON at 4 °C. After washing, cells were incubated with FITC-conjugated anti-mouse (BD Biosciences, San Jose, CA, USA), washed, and mounted with Vectashield mounting medium for fluorescence with DAPI (Vector Laboratories, Burlingame, CA, USA). Images were obtained with an Olympus BX61 microscope, with an attached DP70 Olympus camera and its own software (Olympus, Tokyo, Japan).

### 4.6. Analysis of the Production of ROS and RNS

The presence of ROS/RNS were determined in total cell extracts and supernatants from monolayer cultures of cervical cancer-derived cell lines (C33, SiHa and HeLa), and from PHK transduced with a retroviral vector expressing HPV16 E6 and E7. Briefly, cells supernatants were collected and cleared by centrifugation at 10,000× *g* for 5 min, transferred to fresh tubes and stored at −80 °C until use. For total cell extracts, 10^7^ cells from each culture were resuspended in phosphate buffered saline (PBS) with 0.5% of NP40 (Sigma-Aldrich, St. Louis, MO, USA), sonicated in ice and centrifuged at 10,000× *g* for 5 min. Supernatants were transferred to fresh tubes and stored at −80 °C until use. ROS/RNS were quantified using the OxiSelect™ In Vitro ROS/RNS Assay kit (Cell Biolabs Inc., San Diego, CA, USA), according to the manufacturer’s instructions. Fluorescence emission was measured in a FLUOstar OPTIMA spectrophotometer (BMG Labtech, Ortenberg, Germany).

### 4.7. Analysis of Mitochondrial Membrane Potential (ΔΨ)

Changes in mitochondrial membrane potential (Δ*Ψ*) were determined in cultures of PHK transduced with a retroviral vector expressing HPV16 E6 and E7. Briefly, sub-confluent cultures of cells were treated with 2 nM of TNF as described above. Cells were then detached from the plates, centrifuged, and processed using the BD™ MitoScreen (JC-1) (BD Biosciences, San Jose, CA, USA) kit following the manufacturer’s instructions. Finally, at least 10,000 events per sample were acquired using a FACSCalibur (BD Biosciences, Carlsbad, CA, USA).

### 4.8. Statistical Analysis

Changes of different variables between specific samples pairs were determined using the Student’s *t*-test. Differences were considered as significant when *p* < 0.05.

## Figures and Tables

**Figure 1 ijms-20-00198-f001:**
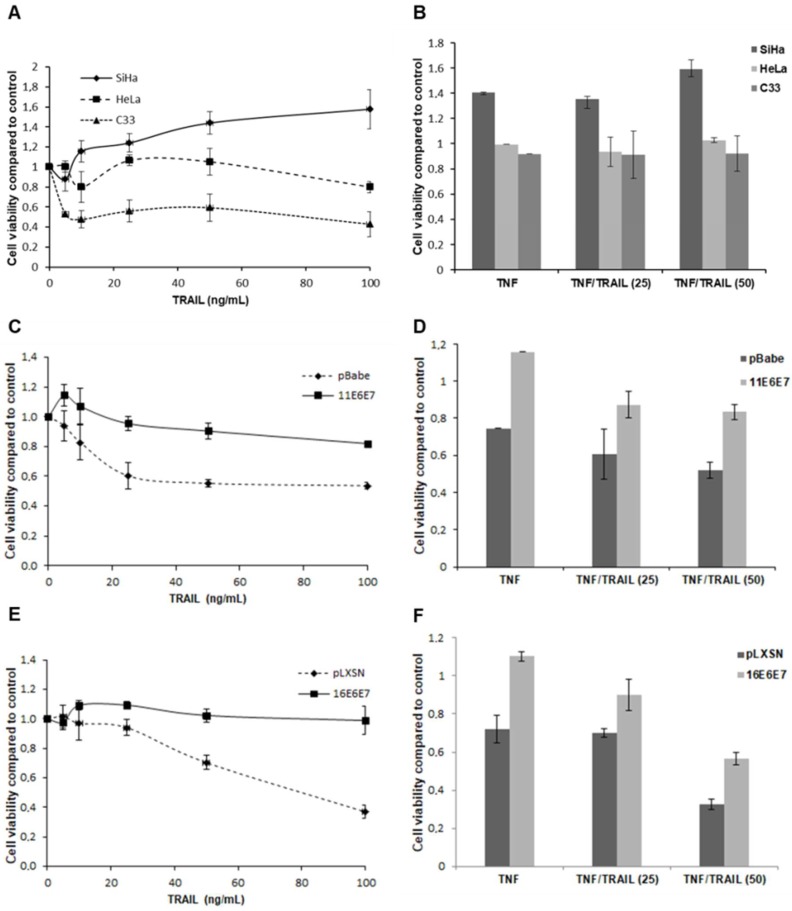
Expression of HPV genes *E6* and *E7* confers resistance to pro-inflammatory cytokines TNF and TRAIL. Effect of TNF and TRAIL on cervical cancer-derived cell lines (**A**,**B**) and PHK transduced with E6 and E7 of HPV11 (**C**,**D**) or HPV16 (**E**,**F**) cell viability. Cells were seeded in 96 wells plates (2000 cells/well) and after 24 h they were treated with 5, 10, 25, 50 or 100 ng/mL of TRAIL (**A**,**C**,**E**), with 2 nM TNF (**B**,**D**,**F**) or with 2 nM TNF plus 25 or 50 ng/mL of TRAIL (**B**,**D**,**F**). All treatments were performed in octuplicate. After 60 h 10 µL of Alamar blue were added per well and cells were incubated at 37 °C in an incubator for at least 4 h. After this period, Alamar Blue’s reduction was monitored in a spectrophotometer through absorbance measurement at 570 e 600 nm. The results presented are representative of three independent experiments. All data are presented as values relative to control.

**Figure 2 ijms-20-00198-f002:**
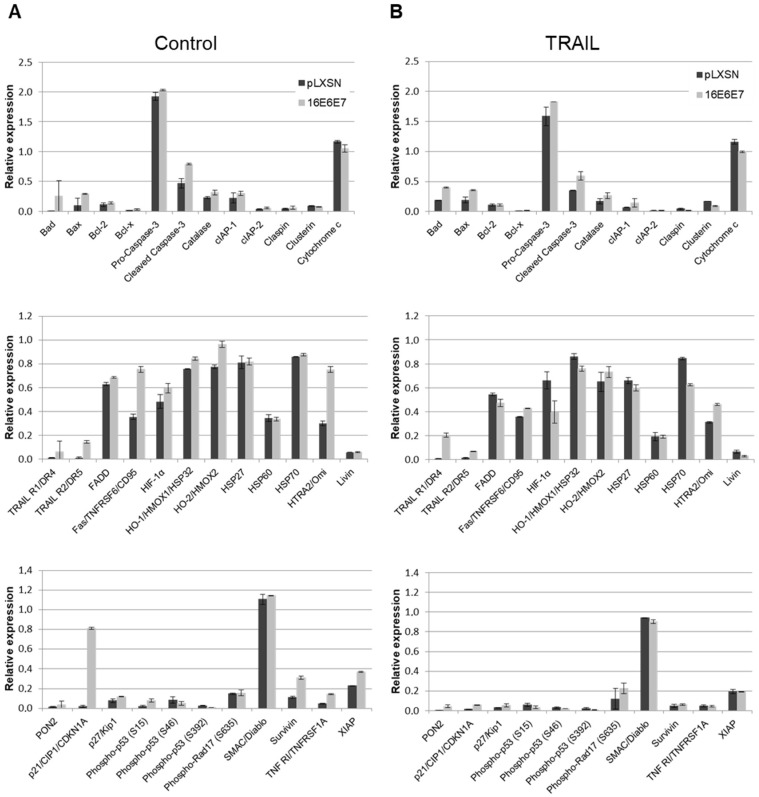
Effect of HPV16 E6 and E7 on the levels of proteins associated with apoptosis regulation/execution upon treatment with TRAIL. The expression of 35 proteins involved in the regulation/execution of apoptosis was determined in total protein extracts of epithelial organotypic cultures established from control (pLXSN) and HPV16 E6/E7 transduced PHKs before (**A**) and after (**B**) treatment with 25 ng/mL of TRAIL. For this purpose, we used the Proteome Profiler™ Array—Human Apoptosis Array Kit (R&D Systems, #ARY009), according to the manufacturer’s instructions. Signals obtained were quantified using ImageJ software and presented as expression relative to endogenous controls. The results presented are representative of three independent experiments.

**Figure 3 ijms-20-00198-f003:**
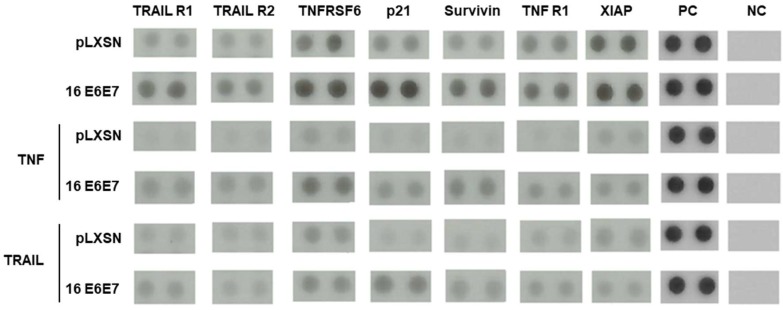
Altered expression of apoptosis regulatory factors in response to TNF and TRAIL in cells expressing HPV16 E6 and E7 oncoproteins. The expression of 35 proteins involved in the regulation/execution of apoptosis was performed as described in Figure 2. Relative proteins signals were quantified using ImageJ software and representative blots of selected differentially expressed proteins are shown. PC: positive control; NC: negative control.

**Figure 4 ijms-20-00198-f004:**
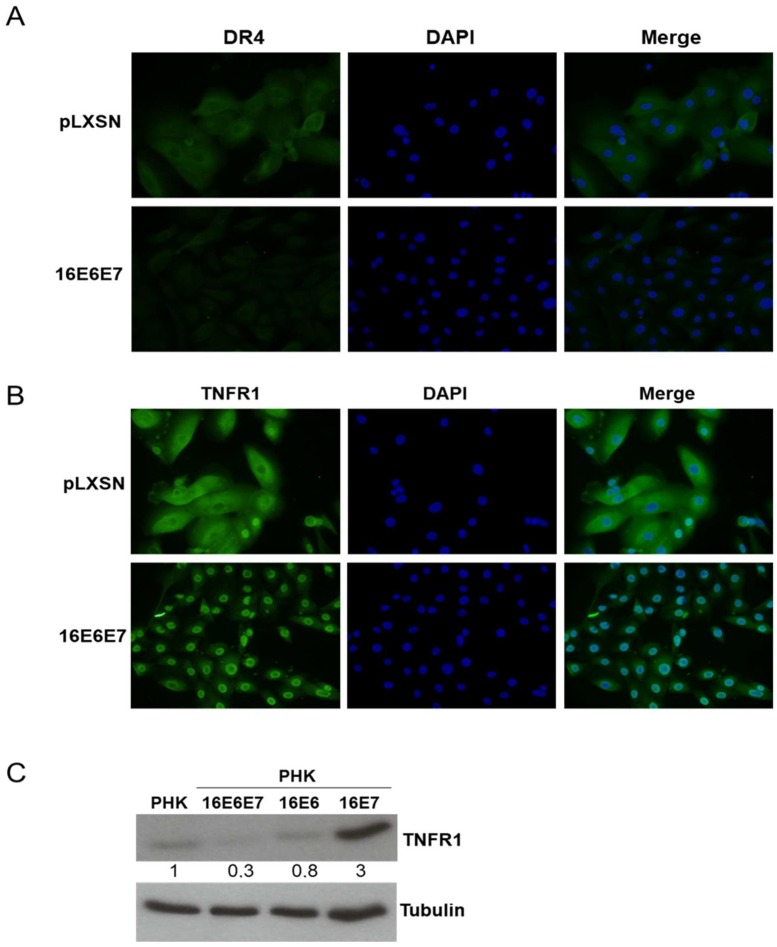
Expression of HPV16E6 and E7 is associated with TNFR1 down-regulation. The expression of DR4 and TNFR1 was determined by immunofluorescence in control (pLXSN) and HPV16 E6/E7 transduced PHKs. Control or HPV16 E6E7 expressing keratinocytes were seeded in 8 wells Lab-Tek chambers, fixed, permeabilized and incubated with anti-DR4 (**A**) or anti-TNFR1 (**B**). After washing, cells were incubated with FITC-conjugated anti-mouse and mounted with Vectashield mounting medium for fluorescence with DAPI. Images were obtained with an Olympus BX61, with attached DP70 Olympus camera and its own software (Olympus, Japan). Magnification: 200×. (**C**) TNFR1 levels were determined by western blot in total protein extracts obtained from the same cells. Relative proteins signals were quantified using ImageJ software.

**Figure 5 ijms-20-00198-f005:**
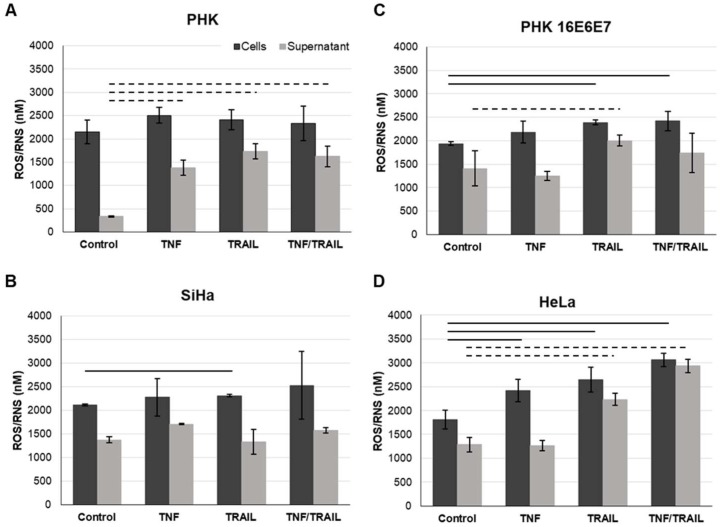
Pro-inflammatory cytokines TNF and TRAIL increase the production and release of reactive oxygen and nitrogen species (ROS/RNS) in cells expressing HPV oncogenes. The levels of ROS and RNS in the absence or presence of pro-inflammatory cytokines alone (2 nM TNF or 50 ng/mL TRAIL) or in combination (2 nM TNF and 50 ng/mL TRAIL) were determined in cultures of PHKs (**A**), PHKs transduced with HPV16 oncogenes (**B**), and in cervical cancer-derived cell lines SiHa (positive for HPV16) (**C**) and HeLa (positive for HPV18) (**D**) using the OxiSelect™ In Vitro ROS/RNS Assay Kit (Cell Biolabs, Inc.), according to the manufacturer’s instructions. The results presented are representative of three independent experiments. Solid lines represent statistical differences between intracellular samples and dashed lines between supernatant samples as determined by Student’s *t*-test (*p*-value ≤ 0.05).

**Figure 6 ijms-20-00198-f006:**
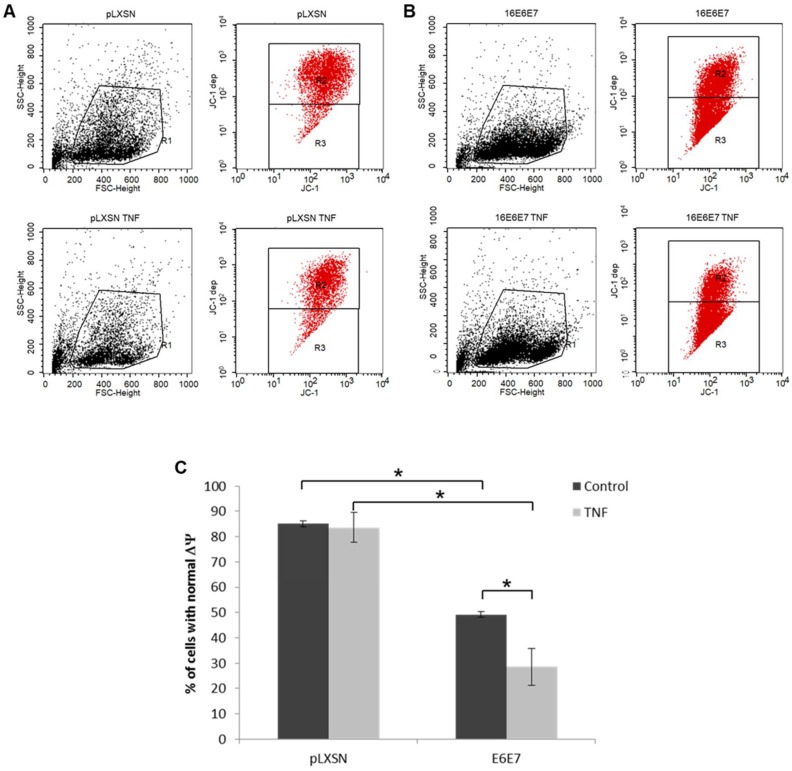
Human keratinocytes expressing HPV16 E6 and E7 exhibit alterations in mitochondria membrane potential (Δ*Ψ*). The status of mitochondria membrane potential was analyzed in monolayer cultures of (**A**) control (pLXSN) and (**B**) HPV16 E6 and E7 expressing PHKs before and after treatment with TNF using the BD™ MitoScreen Flow Cytometry Mitochondrial Membrane Potential Detection Kit according to the manufacturer’s instructions. Mitochondria membrane potential was determined by flow cytometry by measuring fluorescence emission by JC-1 (5,5′,6,6′-tetrachloro-1,1′,3,3′-tetraethylbenzimidazolocarbo-cyanine iodide) in the red channel. This is a cationic cyanine dye that accumulates in mitochondria with preserved Δ*Ψ*. The percentage of cells with normal membrane potential is quantified in (**C**). The results presented are representative of three independent experiments. Statistical differences between samples were determined by Student’s *t*-test (* *p*-value ≤ 0.05).

**Table 1 ijms-20-00198-t001:** Name and function of NFκB-pathway genes differentially expressed between control and HPV16 E6 and E7 expressing cultures (*fold* < −1.5 or > 1.5).

Symbol	Description	*p*-Value	*Fold*
*BCL10*	B-cell CLL/lymphoma 10	0.0047	1.53
*BCL3*	B-cell CLL/lymphoma 3	0.0248	1.70
*NOD1*	Nucleotide-binding oligomerization domain containing 1	0.0079	2.23
*CHUK*	Conserved helix-loop-helix ubiquitous kinase	0.0024	2.86
*CSF3*	Colony stimulating factor 3 (granulocyte)	0.0002	5.81
*ELK1*	ELK1, member of ETS oncogene family	0.0276	1.64
*FOS*	v-fos FBJ murine osteosarcoma viral oncogene homolog	0.0007	−3.18
*GJA1*	Gap junction protein, alpha 1, 43 kDa	0.0018	−3.16
*HTR2B*	5-Hydroxytryptamine (serotonin) receptor 2B	0.0028	2.34
*IKBKB*	Inhibitor of kappa light polypeptide gene enhancer in B-cells, kinase beta	0.0149	2.23
*IL8*	Interleukin 8	0.0002	−1.76
*MYD88*	Myeloid differentiation primary response gene (88)	0.0118	−1.89
*NFKB1*	Nuclear factor of kappa light polypeptide gene enhancer in B-cells 1	0.0060	2.00
*STAT1*	Signal transducer and activator of transcription 1, 91 kDa	0.0361	−2.00
*TBK1*	TANK-binding kinase 1	0.0301	1.77
*TLR3*	Toll-like receptor 3	0.0034	−3.97
*TNFRSF10B*	Tumor necrosis factor receptor superfamily, member 10b	0.0138	−3.20
*TNFSF14*	Tumor necrosis factor (ligand) superfamily, member 14	0.0294	2.13
*RPL13A*	Ribosomal protein L13a	0.0213	1.66

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
