# Peer review of "HPV-Mediated Resistance to TNF and TRAIL Is Characterized by Global Alterations in Apoptosis Regulatory Factors, Dysregulation of Death Receptors, and Induction of ROS/RNS"

_ijms, 2019, doi:10.3390/ijms20010198_

Round 1
Reviewer 1 Report
Herein is shown that combined TNF and TRAIL reduce human primary keratinocytes survival and growth. However, when keratinocytes express the HPV E6 and E7 proteins, they become insensitive to TNF and TRAIL. This is because HPV E6 and E7 can alter cellular expression of survival or pro-apoptosis factors, likely by interfering with the activity of the NFkB transcription factor. As TNF and TRAIL promote the accumulation of the DNA-damaging ROS/RNS inside keratinocytes, these results suggest a mechanism for HPV tumorigenic effect. The present research confirms and expands previous studies by others [Kumari S, Curr Top Microbiol Immunol 2017; Jang Y, Drug Chem Toxicol. 2017; Underbrink MP, J Gen Virol 2016; Nair P, Pathol Oncol Res 1999; Manzo-Merino J, Virology 2014; Zhang W, Sci Rep 2017; Melar-New M, J Virol 2010; Wise-Draper TM, Mol Cell Biol 2006; Thompson DA, Oncogene 2001]. The manuscript is original, and it provides important information to the scientific community.
Author Response
We thank very much Reviewer #1 for the careful reading of our manuscript.
Reviewer 2 Report
The study “HPV-mediated resistance to inflammatory cytokines is characterized by global alterations in apoptosis regulatory factors, dysregulation of death-receptors and induction of ROS/RNS.” gives a contribution on the role of HPV E6/E7 in altering cervical cancer cells and keratinocytes response to Tumor Necrosis Factor (TNF) and TNF-Related Apoptosis-Inducing Ligand (TRAIL).
In general, methodological, scientific parts are accurate and robust coming from a well-consolidated research group. The contribution to existing knowledge on the topic is largely confirmative of data previously obtained by the author’s and by other groups. Anyway, the novelty comes from examining the combined effect of TNF and TRAIL on HPV-transformed cells and human keratinocytes expressing E6 and E7 early proteins from HPV-16 and -11 genotypes; moreover, this study identified new HPV-associated alterations in the apoptosis process and, weakly, also in the induction of ROS/RNS.
The paper is well written but the manuscripts is too long, provided that a more concise text in all sections would help to focus the experimental novelties.
Specific points:
Title: “HPV-mediated resistance to inflammatory cytokines is characterized…..”
the study evaluated only the individual and combined action of the pro-inflammatory cytokines TNF and TRAIL; accordingly, the term “inflammatory cytokines” is too generic. The same holds true for Fig 3 legend.
Abstract:
lane 38: “caused induced” it’s a repetition, one of the two should be eliminated
Introduction:
in the introduction section, there is a detailed presentation of several points besides those examined in this study (e.g caspase activation and apoptosis; E6 and E7 functions), that could be omitted referring to recent reviews in the field.
Methods
lane 175 “107 cells” is probably 107 cells.
Results:
3.4 lanes 299-300 state that “…and TNF R1/TNFRSF1A were up-regulated in cells expressing HPV16 E6 and E7 (Figure 2 and Figure 3)” This appeared as a discrepancy with “the data described above and previously published studies…” so that the authors went in deep and further analyzed (lanes 324-325) “the expression of TNF-receptor 1 (TNFR1) and TRAIL-receptor 1 (TRAIL R1/DR4) by immunofluorescence in keratinocytes expressing HPV16 E6 and E7…”. Below they state that:” TNFR1 down-regulation was confirmed by western blot (Figure 4C). These observations suggest that HR-HPV oncoproteins alter TNFR1 levels…” (lane 331-332).
indeed, WB results are not fully supporting these statements because TNFR1 appears differently regulated in PHK transduced by E6/E7 and E6 with respect to PHK transduced by E7 alone. So this point should be better explained.
3.5 lanes 360-361 and Fig5: it is not completely clear to me, whether and at which extent “…treatment with TNF and/or TRAIL induced higher levels of intracellular ROS/RNS species in HR-HPV-positive cell lines.”. Indeed, Fig5 shows significant differences between each treated cell line and the relative untreated control but there’s no comparison among PHK and the HPV-positive cell lines.
Discussion
lanes 433-443 the authors’ unpublished results and the arguments from the literature are not relevant to this study results that 19 genes related to NFB signaling pathway were differentially regulated in organotypic cultures expressing HPV16 E6 and E7, when compared to controls.
So speculative statements of lanes 433-443 should be removed given the overall length of the paper.
lanes 502-503 and 507-508: these statements are fully supported only whether the authors can better demonstrate that treatment with TNF and/or TRAIL induced higher levels of intracellular ROS/RNS species in HR-HPV-positive cell lines with respect to HPV-negative keratinocytes.
Author Response
First of all, we would like to thank the Reviewer 2 for reading the manuscript so carefully and for the constructive suggestions and questions. We believe the manuscript is much better now, and that is thanks to your contribution.
Point 1: Title: “HPV-mediated resistance to inflammatory cytokines is characterized…..”
the study evaluated only the individual and combined action of the pro-inflammatory cytokines TNF and TRAIL; accordingly, the term “inflammatory cytokines” is too generic. The same holds true for Fig 3 legend.
Response 1: The expression “inflammatory cytokines” was substituted by “TNF and TRAIL” in both the Title of the manuscript and Figure 3 legend.
Point 2: Abstract: lane 38: “caused induced” it’s a repetition, one of the two should be eliminated
Response 2: The word “caused” was eliminated.
Point 3: Introduction: in the introduction section, there is a detailed presentation of several points besides those examined in this study (e.g caspase activation and apoptosis; E6 and E7 functions), that could be omitted referring to recent reviews in the field.
Response 3: We thank very much Reviewer 2 for this suggestion. We believe that the information included in the Introduction is important to contextualize our study. However, in attendance to Reviewer’s 2 comments the following statements were omitted (line numbers and references refer to the original version of the manuscript):
Lines 59-61 “This causes the recruitment and binding of the death-domain-containing adapter proteins TNFR1 associated death domain protein (TRADD) and Fas-associated death domain protein (FADD) [8].”
Lines 80-83 “disturb the activity of several master regulators of cell cycle. The best characterized property of high-risk HPV E6 is its ability to bind p53 and target this protein for degradation [19,20] while E7 inactivates members of the Retinoblastoma protein (pRb) family [21-23]. Furthermore, E6 and E7”
Besides, references 9, 12 and 13 were substituted with “Vanamee ÉS, Faustman DL. Structural principles of tumor necrosis factor superfamily signaling. Sci Signal. 2018 Jan 2;11(511). pii: eaao4910. doi:10.1126/scisignal.aao4910.”
Point 4: Methods lane 175 “107 cells” is probably 107 cells.
Response 4: Thank you for this observation. The number format was corrected.
Point 5: Results: 3.4 lanes 299-300 state that “…and TNF R1/TNFRSF1A were up-regulated in cells expressing HPV16 E6 and E7 (Figure 2 and Figure 3)” This appeared as a discrepancy with “the data described above and previously published studies…” so that the authors went in deep and further analyzed (lanes 324-325) “the expression of TNF-receptor 1 (TNFR1) and TRAIL-receptor 1 (TRAIL R1/DR4) by immunofluorescence in keratinocytes expressing HPV16 E6 and E7…”. Below they state that:” TNFR1 down-regulation was confirmed by western blot (Figure 4C). These observations suggest that HR-HPV oncoproteins alter TNFR1 levels…” (lane 331-332).
indeed, WB results are not fully supporting these statements because TNFR1 appears differently regulated in PHK transduced by E6/E7 and E6 with respect to PHK transduced by E7 alone. So this point should be better explained.
Response 5: We really thank Reviewer 2 for bringing this important point to the Discussion. We have modified some passages of the Results section to address this issue in a more clear fashion and make it clearer to reader. First of all, we have emphasized the fact that we are working with two different systems, monolayer and organotypic cultures that may not produce equivalent results. Besides, we made clear that our results, in each system, are consistent.
The alterations made in the text are:
Lines 299-341: 3.4. HPV16 oncogenes regulate TNFR1 expression in human keratinocytes.
A second important observation made in our study was that the expression of the receptors TRAIL R1/DR4, TRAIL R2/DR5, Fas/TNFSF6/CD95 and TNF R1/TNFRSF1A were up-regulated in epithelial organotypic cultures expressing HPV16 E6 and E7 (Figure 2 and Figure 3). Similar results were obtained after analysis of protein extracts derived from organotypic cultures treated with TNF (supplementary Figure 3) or TRAIL (Figure 2) showing that our observations in this model are consistent. These observations were unexpected considering the data described above and previously published studies addressing the resistance of cells expressing HPV oncogenes grown in monolayer to TNF and TRAIL [6,7]. There are at least two possible explanations for this finding which are not mutually exclusive. The first is that although these receptors are up-regulated, some factor, most probably of viral origin, interferes with its mediated signal transduction. Another possibility is that these receptors exhibit alterations in their cellular localization in the presence of viral proteins.
In order to address this second possibility, we analyzed the expression of TNF-receptor 1 (TNFR1) and TRAIL-receptor 1 (TRAIL R1/DR4) by immunofluorescence in keratinocytes expressing HPV16 E6 and E7. No major differences were observed for TRAIL R1/DR4 levels or localization between control and E6 and E7 expressing cells (Figure 4 A). On the other hand, the cellular distribution and levels of TNFR1 were clearly affected in the presence of HPV proteins. While control cells exhibited a homogeneous distribution pattern all over the cells with areas of concentration at the cellular membrane, cells expressing HPV16 oncoproteins exhibited lower TNFR1 levels located mainly in the perinuclear area (Figure 4B). TNFR1 down-regulation, in the presence of both HPV16 oncogenes, a condition that reflects what it is observed in natural HPV infections, was confirmed by western blot (Figure 4C). These observations suggest that HR-HPV oncoproteins alter TNFR1 levels and induce receptor relocation in cells grown in monolayer contributing to explain, at least in part, the altered response to TNF exhibited by these cells. Besides, the divergent results obtained in monolayer and organotypic cultures indicate that HPV oncoproteins may regulate TNFR1 expression/activity by different mechanisms in different epithelial strata and differentiation conditions (see Discussion).
Point 6: 3.5 lanes 360-361 and Fig5: it is not completely clear to me, whether and at which extent “…treatment with TNF and/or TRAIL induced higher levels of intracellular ROS/RNS species in HR-HPV-positive cell lines.”. Indeed, Fig5 shows significant differences between each treated cell line and the relative untreated control but there’s no comparison among PHK and the HPV-positive cell lines.
Response 6: We again thank Reviewer 2 for this important observation. We agree that the text is not clear. We did not mean that treatment with TNF and/or TRAIL induced higher levels of intracellular ROS/RNS species in HR-HPV-positive cell lines compared to normal PHKs. We meant that PHKs were the only cells that did not exhibit a statistically significant increase in intracellular ROS/RNS upon treatment with TNF and/or TRAIL. In order to explain this point better the text in lines 367-370 was modified as follows “We also observed that single or combined treatment with TNF and/or TRAIL induced a statistically significant increase in the levels of intracellular ROS/RNS species in HR-HPV-positive cell lines, while this was not observed for PHKs in any of the conditions tested (Figure 5, solid lines).”
Point 7: lanes 433-443 the authors’ unpublished results and the arguments from the literature are not relevant to this study results that 19 genes related to NFkB signaling pathway were differentially regulated in organotypic cultures expressing HPV16 E6 and E7, when compared to controls.
So speculative statements of lanes 433-443 should be removed given the overall length of the paper.
Response 7: We believe that the information included in this part of the text is important to highlight the existence of alterations in NFkB signaling pathway in the context of HPV oncogene expression. However, considering the suggestions made by Reviewer 2 we revised this part of the text. The paragraph (lines 426-450 in the original version of the manuscript) was modified it as follows:
Lines 432-448 “The observations described above show that cells expressing HPV exhibit perturbations in both TNF and TRAIL pathways. Therefore, we decided to analyze NFkB signaling pathway since this transcription factor may be activated by both cytokines and regulates a great number of genes [45]. Using a commercial qPCR array we identified 19 genes related to NFkB signaling pathway which mRNA expression was differentially regulated in organotypic cultures expressing HPV16 E6 and E7 when compared to controls. Previous studies have shown that the E7 protein from HR-HPV types reduced TNF-mediated NFkB activation [46]. The effect of NFkB activation in keratinocytes expressing HPV16 oncoproteins is unknown. Several studies indicate that activated NFκB is a negative regulator of keratinocytes proliferation in the epidermis playing an important role in cell differentiation and tissue homeostasis [47-52]. However, it has also been proposed that alterations in NFkB activation or in genes regulated by this factor may play a role in HPV-mediated pathogenesis [53-54]. For instance, it was reported that HPV16 E6 down-regulation by ectopic E2 expression potentiates NFkB activation by TNF in SiHa cells and that this is associated with increased colony formation efficiency [55]. Importantly, our results demonstrate the presence of alterations in the NFkB signaling pathway in organotypic cultures expressing HPV16 oncogenes. These observations support the existence of a complex interaction between HR-HPV oncogenes and TNF/NFkB signal axis relevant to both HPV life cycle and HR-HPV-mediated pathogenesis.
Point 8: lanes 502-503 and 507-508: these statements are fully supported only whether the authors can better demonstrate that treatment with TNF and/or TRAIL induced higher levels of intracellular ROS/RNS species in HR-HPV-positive cell lines with respect to HPV-negative keratinocytes.
Response 8: We addressed this issue when answering to Point 6. Our point here is that the intracellular concentration of ROS/RNS upon treatment with cytokines is upregulated (at least in one condition) in cells expressing HPV oncogenes. On the other hand, single or combined treatment of PHKs with TNF and/or TRAIL did not affect the intracellular concentration of ROS/RNS in any condition tested when compared to untreated cells. The solid lanes in Figure 5 show the statistically significant differences in the content of intracellular ROS/RNS between control and treated cells. Figure 5A, which corresponds to normal PHKs, is the only one that does not show any solid lane.
In order to make this point clear to the reader the text in lines 510-514 of the present version of the manuscript was modified as follows “Here we observed that treatment with TNF or TRAIL induced a statistically significant increase in the intracellular concentration of ROS/RNS in HPV transformed cells as well as in keratinocytes expressing HPV16 oncoproteins (Figure 5B, C and D, solid lines). Interestingly, this was not observed in normal PHKs (Figure 5A).